# A Novel Sensor-Based Application for Home-Based Rehabilitation Can Objectively Measure Postoperative Outcomes following Anterior Cruciate Ligament Reconstruction

**DOI:** 10.3390/jpm13091398

**Published:** 2023-09-19

**Authors:** Natalie Mengis, Sebastian Schmidt, Andree Ellermann, Christian Sobau, Christian Egloff, Mahli Megan Kreher, Korbinian Ksoll, Caroline Schmidt-Lucke, Jules-Nikolaus Rippke

**Affiliations:** 1Department of Orthopedic and Trauma Surgery, Kantonsspital Baselland, 4101 Bruderholz, Switzerland; 2ARCUS Kliniken, Department of Sports Medicine, Rastatter Straße 17-19, 75175 Pforzheim, Germany; sebastian.schmidt@vidia-kliniken.de (S.S.); ellermann@sportklinik.de (A.E.); sobau@sportklinik.de (C.S.); jules-nikolaus.rippke@spitalzofingen.ch (J.-N.R.); 3Department of Orthopedic and Trauma Surgery, University Hospital Basel, Petersgraben 4/Spitalstrasse 21, 4031 Basel, Switzerland; christian.egloff@usb.ch; 4Department of Orthopedic Surgery, Vincentius-Diakonissen-Kliniken gAG, Steinhäuserstraße 18, 76135 Karlsruhe, Germany; 5MEDIACC, Medical-Academic Research Consultancy, 10713 Berlin, Germany; mahli.kreher@mediacc.de (M.M.K.); caroline.schmidt-lucke@mediacc.org (C.S.-L.); 6OPED GmbH, Medizinpark 1, 83626 Valley, Germany; k.ksoll@oped.de; 7Department of Orthopedic and Trauma Surgery, KSA Spital Zofingen, Mühlethalstrasse 27, 4800 Zofingen, Switzerland

**Keywords:** anterior cruciate ligament ACL, telerehabilitation, digital health, digital medical device, inertial measurement sensor, wearable

## Abstract

In order to successfully implement individualized patient rehabilitation and home-based rehabilitation programs, the rehabilitation process should be objectifiable, monitorable and comprehensible. For this purpose, objective measurements are required in addition to subjective measurement tools. Thus, the aim of this prospective, single-center clinical trial is the clinical validation of an objective, digital medical device (DMD) during the rehabilitation after anterior cruciate ligament reconstruction (ACLR) with regards to an internationally accepted measurement tool. Sixty-seven patients planned for primary ACLR (70:30% male–female, aged 25 years [21–32], IKDC-SKF 47 [31–60], Tegner Activity Scale 6 [4–7], Lysholm Score 57 [42–72]) were included and received physical therapy and the DMD after surgery. For clinical validation, combined measures of range of motion (ROM), coordination, strength and agility were assessed using the DMD in addition to patient-reported outcome measures (PROMs) at three and six months after ACLR. Significant correlations were detected for ROM (rs = 0.36–0.46, p < 0.025) and strength/agility via the single-leg vertical jump (rs = 0.43, p = 0.011) and side hop test (rs = 0.37, *p* = 0.042), as well as for coordination via the Y-Balance test (rs = 0.58, *p* ≤ 0.0001) regarding the IKDC-SKF at three months. Additionally, DMD test results for coordination, strength and agility (Y-Balance test (rs = 0.50, *p* = 0.008), side hop test (rs = 0.54, *p* = 0.004) and single-leg vertical jump (rs = 0.44, *p* = 0.018)) correlate significantly with the IKDC-SKF at six months. No adverse events related to the use of the sensor-based application were reported. These findings confirm the clinical validity of a DMD to objectively quantify knee joint function for the first time. This will have further implications for clinical and therapeutic decision making, quality control and monitoring of rehabilitation measures as well as scientific research.

## 1. Introduction

Many patients who have sustained an anterior cruciate ligament (ACL) injury in Germany are advised to undergo ACL reconstruction surgery (ACLR) to restore knee joint function and stability [1,2]. However, post-surgery patients during early rehabilitation are likely to experience functional impediments of the knee such as a decreased range of motion (ROM) due to intraarticular effusion, decreased muscle strength due to an effusion-induced neurogenic inhibition of the quadriceps muscle, pain and higher perceived levels of disability compared to rehabilitation without ACLR. Taken together, these factors can make it difficult to return to daily life and previous activity levels, and may affect the ability to work and exercise [3]. A safe and rapid rehabilitation, resulting in a faster return to daily life and work, has the potential to benefit patients and employers with a positive socioeconomic effect [4].

The rehabilitation process after ACLR typically includes immediate range of motion training, followed by neuromuscular coordination training, as well as strength and speed training [5,6]. The progression between the rehabilitation phases, including the return-to-activity or return-to-sport (RTS) decision, is often still based on time criteria [7]. However, there is a need for an objective, criteria-based rehabilitation progression due to the multifactorial challenges in the process of RTS [7,8]. To plan and control the rehabilitation process, several subjective patient-related outcome measures (PROMs) are used to assess knee function after surgery (e.g., International Knee Documentation Committee Subjective Knee Form (IKDC-SKF) [9], Knee injury and Osteoarthritis Outcome Score (KOOS) [10], Lysholm Score [11], and the Tegner Activity Scale (TAS)). These measures have different levels of practicality and cost-effectiveness. However, only a few objective outcome measures are used in the clinical daily routine (e.g., isokinetic testing for strength [12]). To monitor the improvement in neuromuscular control, objective functional measurements ideally combine measures of speed, agility and dynamic knee valgus moments [13]. Single-leg jumps are known to be sensitive measures, identifying knee function deficits after ACLR, whereas vertical jumps for height appear to be more sensitive in assessing knee function than horizontal hop tests for distance [14].

Furthermore, to date, opportunities to objectively monitor home-based rehabilitation training are lacking and mostly rely on subjective reports from patients. In this setting, it is not yet possible to objectively assess the patient’s rehabilitation progress, the compliance with rehabilitation guidelines or the exercise performance. An objective, easy-to-use measurement tool with the potential to monitor or even manage the rehabilitation progress, to quantify longitudinal follow-up or address scientific issues is missing. The prerequisite for establishing a digital medical device (DMD) is to provide the key quality criteria of validity, reliability and objectivity to enable a highly accurate and precise measurement at a clinically relevant point of time [15]. Additionally, the DMDs need to conform with applicable law and regulations, and be easy and safe to use as well as implemented into a data protection system, such as the ISO 27001 in Germany [16].

With regards to the facts mentioned above, the aim of this study is to assess the clinical validity and reliability of a novel DMD for the objective quantification of knee joint function. Secondary aims were to identify the particular tests that had the highest predictive power in estimating knee joint function after ACLR.

## 2. Materials and Methods

This clinical validation study is a prespecified investigation of an ongoing single-center, prospective, randomized controlled trial (RCT) with the aim to optimize rehabilitation using a sensor-based app after ACLR (DRKS ID: DRKS00024359). The study was conducted in accordance with the Declaration of Helsinki and approved by the local ethics committee (F-2019-048). Patients aged between 18 and 65 years with acute (<6 months between injury and surgery), unilateral ACL tear and the indication for surgical reconstruction were included in the study. Patients were excluded in the case of meniscal bucket handle tear, cartilage damage >ICRS II°, multiligament injury, neurological damage/basic disease, underlying rheumatic disease, previous lower extremity surgery, pregnancy or non-compliance. Written informed consent was obtained from all participants prior to enrolment. For the present analysis, data sets of 67 of 82 patients enrolled into the RCT between July 2019 and December 2020 were included (Figure 1).

The study population reflects the typical young (age 25.3 [21.9–32.0]) and athletic (TAS 6.0 [4.0–7.0]) patients suffering from ACL tears. With a male to female ratio of 7:3 it reflects the average patient population regarding ACL injury. However, if exposed to pivoting or contact sports, women are on average three times more likely to be affected by anterior cruciate ligament injuries than men [17]. In 46 patients (69%), co-injuries, such as damage to the lateral or medial meniscus, cartilage damage (CM2°) or partial medial collateral ligament tear, were reported, reflecting the typical target population for ACLR (Table 1).

Post-ACLR, all patients received a standard therapy plan according to in-hospital standards, adapted to the current state of the art rehabilitation guidelines [6,18]. In the German health care system, a recommendation is issued to the patient on discharge to receive 6 sessions of physiotherapy, which can be extended up to 6 months after the first prescription of physiotherapy. However, there is no national standardized rehabilitation program in Germany. The standard of care includes treatment by the patient’s local physiotherapists with a total of at least twelve sessions consisting of manual therapy or rehabilitation training, depending on individual needs. Patients were instructed on the use of the DMD and the home-based rehabilitation program three days post-ACLR by a specially trained physiotherapist. The rehabilitation program of the DMD is adapted individually to the specific skills, pain and functional abilities of each patient. At first ROM exercises were performed (passive ROM extension/flexion, active ROM extension/flexion), followed by coordination exercises such as the angle reproduction and one leg stance for functional stability of the knee joint. These tests could usually be performed within the first weeks after starting training with the DMD. The one leg squat as well as the Y-Balance test could mostly be performed after 1.5–2 month of coordination and ROM training. The first strength/speed tests to be implemented were the vertical- and distance jump followed by more complex movement sequences such as the side hop, drop jump or speedy hop.

The DMD (Orthelligent^®^, OPED GmbH, Valley, Germany; Figure 2) consists of a small sensor attached to a rubber strap. It is worn at the level of the tibial tuberosity. The sensor itself consists of an inertial measurement unit with a three-axis acceleration sensor and a three-axis gyroscope sensor. The DMD is able to observe signal drifts when a motion variable is determined by integrating a measurement parameter with an unknown or non-specific floating signal offset. It can be used under supervision of an expert but also in a “patient alone” setting using the Orthelligent HOME app [16]. The sensor has to be attached and aligned in such a way that its axis system is oriented along the anatomical axes, i.e., with the *y*-axis parallel to the longitudinal axis of the tibia and the *z*-axis corresponding to the sagittal tibia edge orientation (see Figure 2) [15].

In 2022, Mitternacht et al. demonstrated the technical validity of the device in terms of outcomes provided for the ROM testing with an intrinsic accuracy for angle measurements from 1 to a maximum of 2 degrees for the DMD device in comparison to video-motion analysis, as well as for the vertical jump, side hop and speedy hop tests with almost identical timing [15]. The app, which is provided in combination with the DMD, evaluates test performance and can provide instant visual feedback. The instant feedback assists patients and healthcare providers to determine the rehabilitation status and monitor the rehabilitation progress. The so-called “FIT-Index” (identical to the limb symmetry index (LSI)), calculated by taking the average of any test scores for the affected limb, divided by the unaffected limb, multiplied by 100 to obtain a percentage difference between limbs, is calculated by the app and used for intraindividual comparison of the injured with the non-injured extremity [16,19]. Furthermore, the progress of the rehabilitation can be assessed by comparing the results with those of previous controls. The app was downloaded on smartphones and tablets for iOS and Android systems and included instructional videos to guide patients through the movement tests and exercises. The test battery of the DMD includes different ROM, coordination, strength and agility tests, which are assigned to different phases of rehabilitation. In total, twelve tests could be performed. A summary of the available tests and corresponding outcome parameters is provided in Table 2. All tests, and therefore test values included in this analysis, were performed under clinical guidance and supervision by a specialist (orthopaedic surgeon or physiotherapist) according to manufacturer instructions for the injured and non-injured limb. If a patient was unable to perform a specific jump or DMD test (e.g., due to lack of confidence in the stabilization of the knee joint), this specific jump or DMD test that was not performed was not included in the analysis, but the rest of the DMD test results could still be analyzed. All patients that accepted the follow-up examination in the study center had their jump values recorded. Patients were allowed to perform three repetitions per test and the best was recorded as the test result. This approach was previously agreed upon after analysis of previous data from 2 independent trials [16,20,21].

Patients were asked to complete the IKDC-SKF, TAS and Lysholm score before surgery (T_0_) as well as at three (FU1) and six (FU2) month follow-ups. At FU1 and FU2, patients additionally underwent a physical examination after completing the questionnaires. Some patients did not complete the PROMs or did not fulfill all questions needed within the questionnaires, which led in parts to reduced numbers of patients for statistical analysis (Figure 2). Due to the selected intention-to-treat approach, these patients were not regarded as dropouts since they continued with the ongoing trial.

All continuous variables were examined for normal distribution with the Kolmogorov–Smirnov test. Normally distributed data are presented as mean ± standard deviation (MW ± SD), while non-normally distributed data (demographic data and results of DMD tests) are presented as median with corresponding quartiles (median [Q1–Q3]) and were compared by the Wilcoxon test for two dependent variables or by the Friedmann test with more than two dependent variables. *p*-values were adjusted for multiple comparisons according to the Bonferroni–Holm method. Spearman’s correlation coefficients were calculated for PROMs in order to reveal internal consistency and for outcome parameters of the DMD tests and the IKDC-SKF at three and six months, respectively. Furthermore, multivariate regression analysis (ANOVA, stepwise) was performed to detect independent predictors of the IKDC-SKF score at three and six months, respectively. Explorative correlation analyses were assessed for outcome parameters of the DMD and further knee function and physical activity measures. Statistical analysis was performed by using SPSS 28 (IBM, Armonk, NY, USA). The level of statistical significance was set at *p* < 0.05.

Additional to the clinical validation of the DMD, the minimally clinical important difference (MCID) for the vertical jump test was calculated, which has been proven to be a suitable test in determining knee function [14]. The MCID is defined as the smallest difference of a score indicating clinical patient-relevant changes. For the IKDC-SKF, a change of more than 10.7 units was reported as the MCID in a Japanese patient population [22]. The MCID was calculated for the vertical jump using the threshold value for the IKDC-SKF of 10.7 units at 3 months follow-up. The difference in the IKDC-SKF at 3 and 6 months postoperative was dichotomized into “improved” and “not improved” for each patient. For this binary outcome of the IKDC-SKF, a receiver operating characteristic (ROC) curve was generated using different cut-off values. The MCID of 2 cm for the vertical jump test was determined as the cut-off value that optimizes both sensitivity and specificity according to the Youden index that optimizes sensitivity and specificity equally [23].

## 3. Results

### 3.1. Patient-Reported Outcome Measures and Functional Assessment

The preoperative functional status (T_0_, baseline) of the patients assessed by the IKDC-SKF (47.1 [30.5–59.8) and the Lysholm Score (56.5 [42.3–72.3]) show a reduced level of function with regard to the functionality of an uninjured knee joint. Compared with the preoperative value, the IKDC-SKF increased significantly by 36.7% at 3 months, FU1 (64.4 [51.7–73.6], *p* < 0.001), and by 66.0% at 6 months, FU2 (78.2 [71.9–83.4], *p* < 0.001; Figure 3).

Likewise, a significant improvement in the Lysholm Score of 39.8% at 3 months and 54.0% at 6 months was observed (*p* < 0.001 respectively). Before ACLR, the TAS, which measures the patients’ sportive level prior to the injury, was above average. Patients scored 6.0 [4.0–7.0] points with a reduction of 33.3% after 3 months and subsequent increase by 25% at 6 months (*p* < 0.001 respectively). A summary of the PROMs describing knee joint function and physical activity is provided in Table 3.

Spearman correlation analysis revealed good internal consistency for all PROMs assessing the functional knee status. Significant correlations were detected between the IKDC-SKF and Lysholm Score preoperatively (r_s_ = 0.81, *p* < 0.01), at three (r_s_ = 0.59, *p* < 0.01) and at 6 months (r_s_ = 0.65, *p* < 0.01). Likewise, there were significant correlations between the IKDC-SKF and TAS at three (r_s_ = 0.60, *p* < 0.01) and six months (r_s_ = 0.67, *p* < 0.01), whereas no significant correlations were found preoperatively.

After three months, significant correlations were found for the IKDC-SKF score, active (r_s_ = 0.46, *p* = 0.0004, y = 52.52 + 0.58x, n = 38) and passive ROM (r_s_ = 0.36, *p* = 0.025, y = 93.34 + 0.41x, n = 39). Further, significant correlations with the IKDC-SKF score were found for strength via the vertical jump test (r_s_ = 0.43, *p* = 0.011, y = −41.99 + 2.78x, n = 35), agility via the side hop test (r_s_ = 0.37, *p* = 0.042, y = 3.66 + 0.58x, n = 31) and for coordination via the Y-Balance test (r_s_ = 0.58, *p* ≤ 0.0001, y = 32.04 + 1.08x, n = 38; Figure 4).

Regarding the secondary endpoint at six months, significant correlations were found between the IKDC-SKF and the vertical jump test (r_s_ = 0.44, *p* = 0.018, y = −0.014+ 3.95x, n = 28), side hop test (r_s_ = 0.54, *p* = 0.004, y = −40.52 + 1.12x, n = 27) and Y-Balance test (r_s_ = 0.50, *p* = 0.008, y = 25.27 + 0.95x, n = 27; Figure 5). Moreover, the vertical jump test (ß = 0.48, T = 2.9) and passive ROM (ß = 0.35, T = 2.2) were independently correlated with the IKDC-SKF score at 3 months (*p* = 0.003) and the side hop test (ß = 0.45, T = 2.5, *p* = 0.022) at 6 months.

Patients that improved in the IKDC-SKF higher than the prespecified MCID of 10.7 units (n = 16) had a higher mean increase in vertical jump height from 3 to 6 months (7.5 cm) compared to those with a defined but not clinically relevant improved IKDC-SKF (*p* = 0.014). The MCID for the vertical jump at early to mid-stage rehabilitation was calculated as 2 cm (area under the ROC curve (AUC; Figure 6): 0.71 (95% CI [0.49–0.94]); this corresponds with a sensitivity of 81% (95% CI: 54–96%) and a specificity of 70% (95% CI [35–93%]). The Youden index at the cut-off value was 0.512 based on 28 patients who completed the IKDC-SKF and the vertical jump test at FU2.

### 3.2. Explorative Correlation Analysis between DMD Tests TAS and Lysholm Score at Three and Six Months

The initial TAS correlated with the respective individual TAS at 3 and 6 months (r_s_ = 0.39, *p* = 0.016 and r_s_ = 0.58, *p* = 0.001, respectively). Furthermore, a significant correlation was detected between the TAS and vertical jump test at three months (r_s_ = 0.40, *p* = 0.024). Significant correlations were detected between the TAS and side hop test (r_s_ = 0.62, *p* < 0.001), vertical jump test (r_s_ = 0.67, *p* < 0.001), distance jump test (r_s_ = 0.55, *p* = 0.003), one leg squat test (r_s_ = −0.45, *p* = 0.022) and speedy jump (r_s_ = −0.45, *p* = 0.025) at six months. At three months, a significant correlation was detected between the Lysholm Score and the active ROM test (r_s_ = 0.33, *p* = 0.043). However, no significant correlations were found between the Lysholm Score and DMD tests at six months.

### 3.3. Adverse Events and Complications

During the course of the study, five of 67 included participants reported six adverse events that were unrelated to the DMD’s use (Table 4).

## 4. Discussion

The aim of the study was the clinical validation of a novel and easy-to-use DMD for quantifiable and objective assessment of knee joint function. For this, the outcome of the DMD was compared to validated subjective assessment scores within only the early and mid-stage rehabilitation phases after ACLR. The DMD has already been validated regarding its accuracy to measure timing-related jump tests as well as angle measurements in healthy individuals under a controlled laboratory environment [15]. This study can close the gap between laboratory and clinical settings. The results show a strong correlation between the DMD test outcomes and the subjective knee function. Thus, the DMD gives therapists the opportunity to objectify and quantify the rehabilitation process of their patients.

The subjective IKDC-SKF demonstrated significant as well as clinically relevant improvements three and six months after ACLR. This is consistent with previous longitudinal studies that implemented the IKDC-SKF for assessing knee function in an ACLR population [24,25]. Assessing symptoms, activities of daily living and sport activities [9], the validated IKDC-SKF is qualified to detect correlations between the PROM and the objective measurements obtained with the DMD. This has been confirmed by the positive correlation demonstrated for the IKDC-SKF and the DMD tests’ active and passive ROM, vertical jump and side hop test, as well as the Y-Balance test at three months. Thus, improvements in the objective criteria of ROM, strength and coordination are also predictive of improvements in subjective knee function during early and mid-stage rehabilitation and vice versa. Interestingly, two functional tests (vertical jump, side hop) were found to independently correlate with the subjective knee function, apparently assessing different knee functional components. Therefore, the MCID of the vertical jump test was calculated. An increase in individual vertical jump heights higher than 2 cm might be considered as relevant when assessing knee functional status in early to mid-stage rehabilitation. Similar to the IKDC-SKF, analysis of the Lysholm Score demonstrated significant improvements over time, probably due to the ongoing healing process [26]. Although the Lysholm Score includes the domains of symptoms, complaints and activities of daily living, the sport activity level is not included [25]. Thus, the observed correlation with the DMD test for active ROM at three months might reflect the decreased swelling and reduced pain of the injured knee in the early rehabilitation phase. The absence of correlations for strength and coordination tests might also be due to the missing sport activity domain in the Lysholm Score. This emphasizes the external validity of the DMD’s functional tests.

Observed changes in TAS levels over time do not reflect consistent improvements between preoperative values and values after three and six months. However, this was to be expected as previous investigations on return to preinjury sports level post-ACLR have shown that only about 65% of athletes return to their preinjury sports levels, even in long-term follow-up. Furthermore, athletes who are not professionally engaged in sports are less likely to return to preinjury levels [27]. The cohort study of Kim et al. (2021) investigated 91 recreational athletes 12 months after ACLR and reported a similar development of TAS levels. Participants experienced a decrease in TAS levels after ACLR. During rehabilitation, the participants´ activity increased at 6 months and at 12 months. However, the preinjury TAS level was not reached [28]. Analogous results could be demonstrated for patients with complex patellofemoral disorders [29]. This suggests that the healing processes might be a dominant factor in the early rehabilitation phase, while sportive aspects, necessary to reach higher scores in TAS, are less dominant in the early rehabilitation phase. The correlation observed between the 3-month mean TAS and the vertical jump performance of the DMD test, and even more so, the correlations at 6 months between the TAS and five dynamic tests of the DMD (side hop test, vertical jump test, distance jump test, one leg squat test and speedy jump) reflect the validity of the DMD assessment, as these tests are related to athletic performance. It further outlines the relevance of the vertical jump test since this has been shown to be the first test to be sensitive to sportive activity changes already in the early rehabilitation phase. However, the correlation of the DMD test outcomes and TAS was not defined as the primary endpoint because it only reflected the participant’s level of physical activity, while the IKDC-SKF questionnaire assessed a wider range of factors, including symptoms, function and daily activities. The safety of the DMD for use in a home-based setting was confirmed within this clinical validation study, as none of the observed (serious) adverse events were attributed to the DMD. The 4% incidence of arthrofibroses within this cohort corresponds with reported rates of arthrofibroses of roughly 5% [30]. Furthermore, the incidence of partial ACL rupture post-ACLR within this present study of 1.5% is lower than the expected rate of more than 5% [31].

To optimize the process of rehabilitation, repetitive testing and monitoring of patient’s abilities would be ideal. In everyday clinical practice, this, however, poses a considerable burden because clinicians and physiotherapist’s capacities to assess their patient’s functional abilities following ACLR are limited. The use of DMDs may overcome these limitations as validated and reliable tools to support rehabilitation monitoring in clinical practice and enables patients’ individualized at-home rehabilitation training.

Still, there are a few limitations in this study. The present analysis only included a time course of six months where a full functional recovery is not expected. Further studies investigating the long-term follow-up after ACLR of at least 2 years should be pursued to consolidate the evidence presented in this paper. The precise and free-of-play attachment of the sensor is crucial, especially the exact frontal alignment to the tibia edge. The adjacent body segment relative to which angular motion is determined must maintain a fixed or known position and orientation during the observed motion sequences [15]. This may pose a serious hurdle to the accuracy of testing, especially in a home setting. During the DMD-assisted rehabilitation program, the test requirements and the number of tests performed increase. However, the number of tests as well as the time of first usage of test categories (ROM, coordination, strength/speed) varies greatly from individual to individual [16]. Since the functional tests were only completed when the patients felt ready for test execution, the number of patients during the follow-ups is reduced. In particular, the more complex movement tasks with a high demand of neuromuscular control and strength were likely not performed within the first 6 months after ACLR. Better DMD test results are expected at twelve months, or later. Furthermore, the association between skeletal muscle dysfunction and subjective knee disability is poorly understood. Previous studies reported varying degrees of association between objective and subjective outcomes of knee function [3,32]. The inconsistencies in the literature may be due to differences in populations, pathologic condition and testing methodology and equipment. The population included in this clinical validation study represents the typical, athletic ACLR patients, as evidenced by mean age, BMI, co-injuries and TAS. Although the transferability of study results was attempted by using the typical population as well as common subjective scores and objective movement tasks, care must be taken when determining the MCID of the vertical jump test. The study populations that were used determining the MCID of the IKDC-SKF (Japanese population, [22]) and the vertical jump (German population) differ regarding ethnicity and existing health care system, and thus, the transfer of the study’s results might be influenced.

## 5. Conclusions

This study demonstrated the clinical validity and safety of a novel DMD to be implemented as an objective quantitative measurement instrument in the early and mid-stage rehabilitation phases after ACLR. The DMD allows objective standardized quantification of simple and complex dynamic knee function for physicians and patients in clinical practice and research. Furthermore, the one leg vertical jump and the side hop test measured by the DMD seem to be excellent indicators of functional knee recovery within the early and mid-stage rehabilitation phase, as significant correlations with the IKDC-SKF were demonstrated at different timepoints. A future investigation should consider a long-term follow-up period of up to two years.

## Figures and Tables

**Figure 1 jpm-13-01398-f001:**
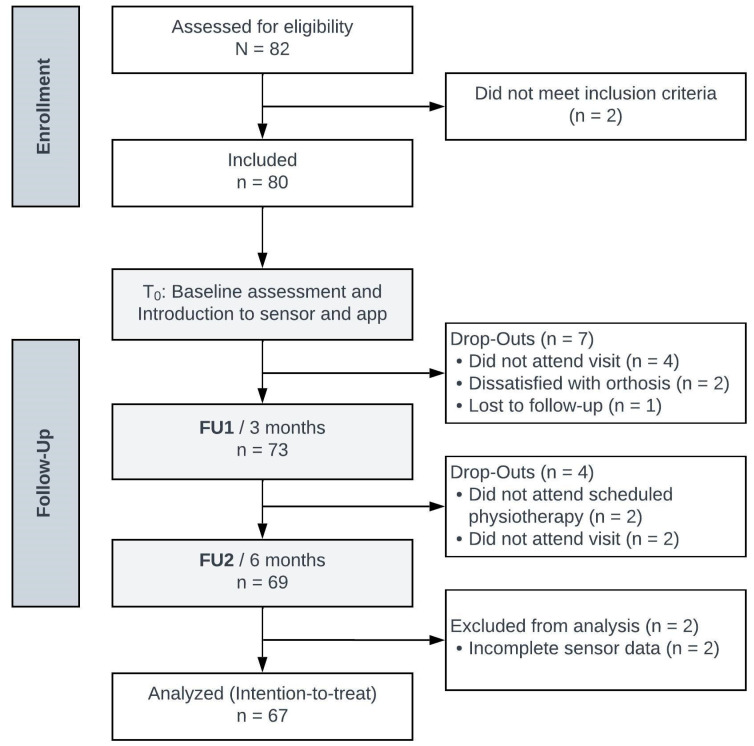
Participant study flow diagram adapted from the CONSORT RCT flow diagram (n, number of participants; T0, baseline; FU, follow-up).

**Figure 2 jpm-13-01398-f002:**
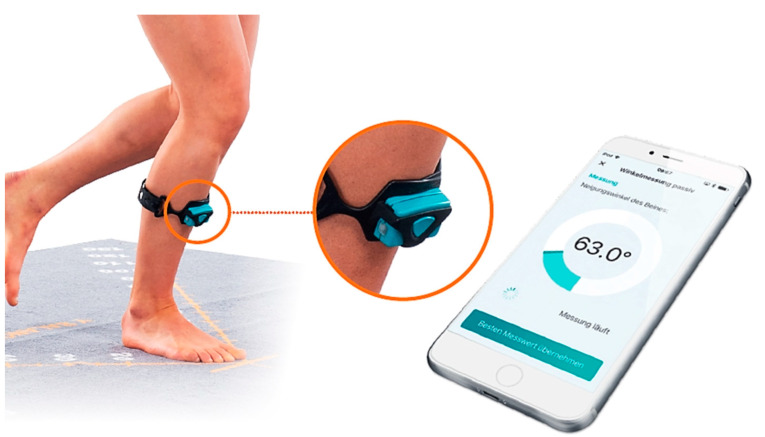
Presentation of DMD (Orthelligent^®^) and medical application.

**Figure 3 jpm-13-01398-f003:**
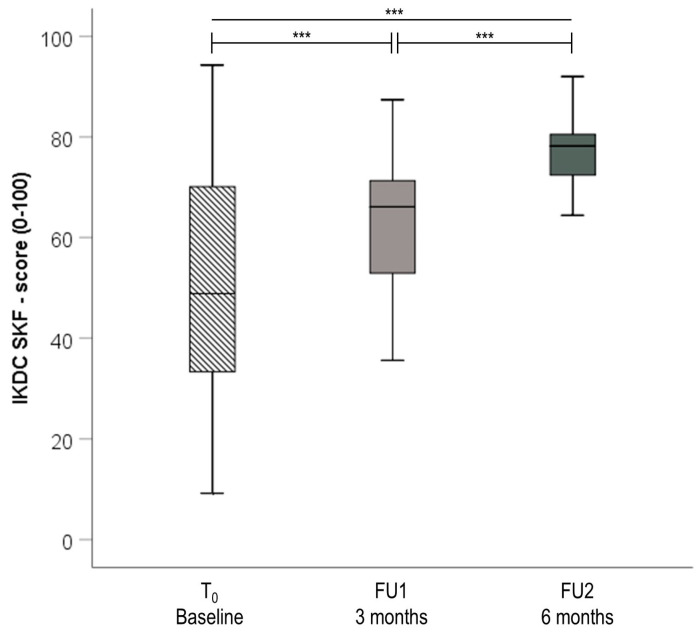
Box plots for differences in IKDC-SKF preoperatively (T_0_, baseline), 3 and 6 months (IKDC-SKF Score (0–100): 100 points indicate highest level of function and lowest level of symptoms. Significance is indicated by *** *p* < 0.001; Abbreviations: IKDC-SKF, International Knee Documentation Committee Subjective Knee Form; n, number; T0, baseline; FU, follow-up).

**Figure 4 jpm-13-01398-f004:**
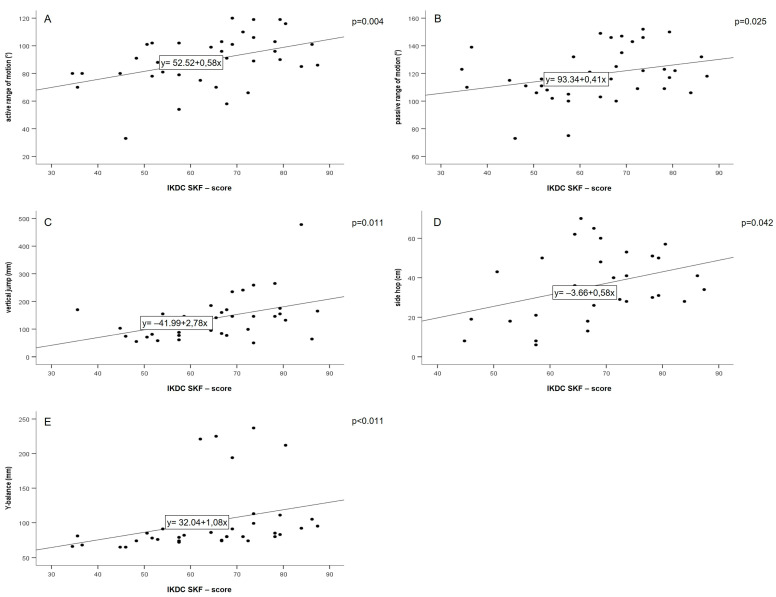
Spearman correlation plots for active range of motion (**A**), passive range of motion (**B**), vertical jump (**C**), side hop (**D**), Y-balance, and IKDC-SKF scores at three months (**E**), (FU1; abbreviations: IKDC-SKF: International Knee Documentation Committee Subjective Knee Form).

**Figure 5 jpm-13-01398-f005:**
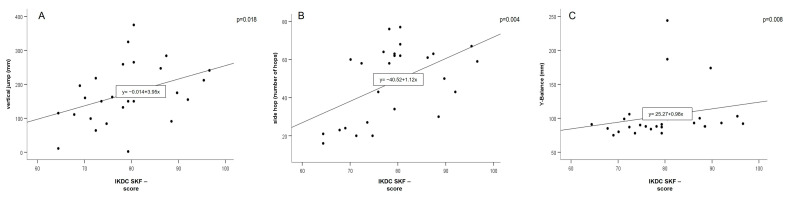
Spearman correlation plots for vertical jump (**A**), side hop (**B**), Y-balance (**C**), and IKDC-SKF scores at six months (FU2; Abbreviations: IKDC-SKF, International Knee Documentation Committee Subjective Knee Form).

**Figure 6 jpm-13-01398-f006:**
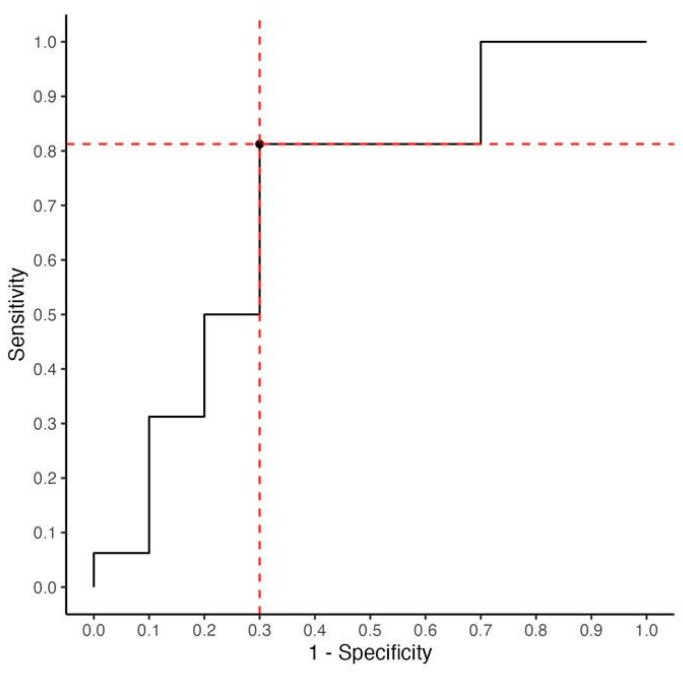
ROC curve for determining the MCID of the vertical jump test. The MCID of 2 cm optimized sensitivity (=81%; 95% CI [54–96%]) and specificity (=70%; 95% [CI: 35–93%]). The optimized threshold of 2 cm (highest specificity and sensitivity) is illustrated as intersection of the red dotted lines.

**Table 1 jpm-13-01398-t001:** Demographic data.

Characteristics (n = 67)	n (%)
**Sex**	
Male	47 (70.1%)
Female	20 (29.9%)
	**Median [Q1–Q3]**
Age [years] ^1^	25.3 [21.9–32.0]
Height [cm]	177.0 [169.0–183.0]
Weight [kg]	75.5 [67.3–85.0]
BMI [kg/m^2^]	23.9 [22.5–26.3]
Tegner Activity Scale, T_0_	6.0 [4.0–7.0]
Lysholm Score, T_0_	56.5 [42.3–72.3]
IKDC-SKF Score, T_0_	47.1 [30.5–59.8]
	**n (%)**
**Injured Leg**	
Right	27 (41%)
Left	40 (59%)
	**n (%)**
**Diagnosis for acute co-injuries**	
ACL rupture—ACL reconstruction (Semitendinosus/Gracilis)	67 (100%)
Lateral meniscus tear, repair	10 (15%)
Lateral meniscus tear, partial meniscectomy	5 (8%)
Medial meniscus tear, repair	12 (18%)
Medial meniscus tear, partial meniscectomy	2 (3%)
Cartilage damage (CM2°), chondroplasty	4 (6%)
Partial medial collateral ligament tear, non-operative	7 (10%)
Adhesions arthrolysis	4 (6%)
Hematoma flushing	1 (1%)
Lateral collateral ligament strain, non-operative	1 (1%)

^1^ range of included patients 18–57 years.

**Table 2 jpm-13-01398-t002:** Digital medical device tests and outcome parameters.

Category	Test	Outcome Parameter	Optimal and Earliest Timepoint for Assessment Postoperatively *
Range of Motion	Angle measurement passive	Flexion (degrees), extension (degrees)	3–5 days
	Angle measurement active	Flexion (degrees), extension (degrees)	3–5 days
	Extension deficit	Extension deficit (degrees)	3–5 days
Coordination	One leg squat	Knee displacement (degrees)	1–1.5 months
	Angle reproduction	Deviation from predefined angle (degrees)	1–2 weeks
	Y-Balance	Dynamic Balance Composite Score (mm)	1–1.5 months post
	One leg stance (Functional stability)	Knee displacement (mm)	2–3 weeks
Strength/Speed	Vertical jump	Jump height (mm)	1–1.5 months
	Distance jump	Jumping distance (cm)	1.5–2 months
	Side hop	Number of hops performed within 30 s	2–3 months
	Drop jump	Knee displacement (degrees)	2–3 months
	Speedy jump	Time (s) to complete small obstacle course	2–3 months

* All timepoints are based on current clinical expert consensus [6]. Abbreviations: s, seconds.

**Table 3 jpm-13-01398-t003:** Patient-reported outcome measures on knee joint function and physical activity.

	IKDC-SKF Score	Lysholm Score	Tegner Activity Scale
**T_0_**	47.1 [30.5–59.8]	56.5 [42.3–72.3]	6.0 [4.0–7.0]
**FU1**	64.4 [51.7–73.6] ***	79.0 [63.5–85.5] ***	4.0 [3.0–4.0] ***
**FU2**	78.2 [71.9–83.4] ***	87.0 [80.0–90.0] **	5.0 [4.0–5.0] **

** *p* < 0.01 and *** *p* < 0.001 indicates sig. difference to baseline (T_0_-FU1) and in follow-up (FU1-FU2). Abbreviations: IKDC-SKF, International Knee Documentation Committee Subjective Knee Form; T_0_, baseline; FU, follow-up.

**Table 4 jpm-13-01398-t004:** Summary of adverse events.

Type of Event	Description	Total (n = 67)
Serious adverse event		
	knee joint infection with surgical irrigation, ACL preserved	1
Adverse event		
	patellofemoral pain syndrome	1
	ACL plastic partial rupture, no surgery required	1
	Arthrofibrosis with indication for oral cortisone therapy	2
	Arthrofibrosis after infection	1

Abbreviations: n, number of patients; ACL, anterior cruciate ligament.

## Data Availability

All data, tables and figures presented in this manuscript are original. Further inquiries can be directed to the corresponding author.

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
