# Peer review of "A Novel Sensor-Based Application for Home-Based Rehabilitation Can Objectively Measure Postoperative Outcomes following Anterior Cruciate Ligament Reconstruction"

_jpm, 2023, doi:10.3390/jpm13091398_

Round 1
Reviewer 1 Report
In this clinical study the short-term results of ACL reconstruction were studied and the use of an inertial device was explored.
The study has several serious methodological problems though:
1. The follow up period is very short (6 months) and the recovery of the patients is still incomplete
2. The rehabilitation program is not described explicitly and is probably not the same in all patients.
3. The test-test reliability of the measurements carried out by the DM is not reported.
4. The positioning of the DMD at the level of the tibial tuberosity may be subjected to significant positional variation, and this may affect the accuracy of the results.
5. The FIT index is not accurately described.
6. The outcome of the 3 tests used is not accurately described and not reported in numbers, but rather only the correlations are presented.
Keywords: please check again the keywords telerehabilitation and wearable
Lines 20 and 21: please restructure the sentencing
Line 44: Most patients who have sustained an anterior cruciate ligament (ACL) injury, are 44 advised to undergo ACL reconstruction surgery. This statement is not valid.
Line 46: these patients are likely to experience functional impediments. This statement is not correct. Complications and patient dissatisfaction is expected but it is not likely to occur.
Line 48. and higher perceived levels of disability. This is not correct.
Line 63. clinical practicability and economic efficiency. Please rephrase
Line 63-64. However, only few objective outcome measures are used. This is not absolutely correct. There are various objective outcome measures in use.
Line 90.” Patients aged between 18 and 65 years with acute (< 6 months between injury and surgery).” Please report how many middle-aged patients did you include in your study. In Table 2 you report that the median age was 25.3 years [21.9-32.0]. Additionally, the acute period for ACL reconstruction is 3-6 weeks but not longer. In your study you treated patients with chronic ACL tears with various intraarticular injuries.
Line 92. “Patients were excluded in case of meniscal bucket handle tear, cartilage damage >ICRS II°.” You included patients with meniscus repair and excision. Why did you specifically excluded patients with a bucket handle tear, which is repairable. The same applies with cartilage lesions.
Line 103. Please provide more information about the rehabilitation program. Do you think that 12 more or less sessions are sufficient to rehabilitate a patient with ACLR?
Lines 133-137. “If a patient was unable to perform a test (e.g., due to lack of confidence in the stabilization of the knee joint), no result was recorded at the follow-ups and was regarded as missing value within this analysis accordingly.” This is not really a valid methodology. Failure to complete the test is not a missing value.
Line 137. “Patients were allowed to perform three repetitions per test and the best was recorded as the test result”. The mean value would be more valid to use.
Line 142. “Some patients denied the completion of the PROMs or did not fulfill all questions needed.” This is strange. Why did this happen? A satisfied patient will participate in the study.
Line 214. The correlations of the vertical jump test, the side-hop test, the Y-Balance test and the passive ROM are reported but not the actual results of the evaluation.
Figures 4 and 5. The y axis description is not readable.
Line 245. The 5 patients with a severe adverse effect were included in the final evaluation?
The discussion is rather extensive and should be focused on the findings of the study.
The limitation section is not sufficient.
minor corrections are necessary.
Reviewer 2 Report
In this manuscript, the authors demonstrated the clinical validiaty and safety of a novel DMD to be implemented as an objective measurement instrument in the early and mid-stage rehabilitation phases after ACLR. Overall, the article seems very interesting.
I present my comments and suggestions for changes in relation to the following parts of the article.
- Table 2 shows the statistical information of the participants in the study. Typically, these information is described in the Methods section.
- The male to female ratio is shown as 7:3. In general, there is a difference between male and female athletic performance. Didn't this ratio affect the results? Looking at Figures 4 and 5, some results show a wide distribution of data. Considering these points, it would be better to analyze the results with only 70% of men. Please describe this in the discussion section.
- In this study, the clinical efficacy of DMD with home-based rehabilitation was evaluated. Of course, as described in the text, I don't think there is any difficulty in using it if you receive training from an expert. However, is it possible for a patient alone to accurately perform 12 tests using the device? It is considered that if the measurement method or environment is not the same, it may affect data accuracy.
- It seems that the unit of the Outcome parameter on the y-axis in Figure 4 is different from Table 1. Please check and correct.
- Figure 4 and Figure 5 are unclear and difficult to understand. Please insert a figures with improved resolution.
- In the text, sensitivity was described as 81% and specificity as 70%. Please modify Figure 6 to fit the description.
- The results of TAS are rather poor compared to other test results (IKDC-SKF score, Lysholm-Score). What does this mean? In addition, it was described that there was a significant correlation between DMD test and TAS results. I think it will be helpful to understand if tables or figures are added.
Overall well written.
I think only minor English corrections are needed.
Round 2
Reviewer 1 Report
Although the paper has several methodological weaknesses and not all conclusion can be generalised, the methodology and the effort of the authors should be sufficient to allow publication of their paper.
Reviewer 2 Report
I think it is well done, thereby making it easy to understand.
Congratulations for the work done.